# The nanoscale organization of the Wnt signaling integrator Dishevelled in the vegetal cortex domain of an egg and early embryo

**John H. Henson** [1,2] *, **Bakary Samasa** [1,2], **Charles B. Shuster** [2,3], **Athula H. Wikramanayake** [4]

**1** Department of Biology, Dickinson College, Carlisle, Pennsylvania, United States of America, **2** Friday Harbor Laboratories, University of Washington, Friday Harbor, Washington, United States of America, **3** Department of Biology, New Mexico State University, Las Cruces, New Mexico, United States of America, **4** Department of Biology, University of Miami, Coral Gables, FL, United States of America

* henson@dickinson.edu

**Data Availability Statement:** All relevant data are within the manuscript and its Supporting Information files.

## Abstract

Canonical Wnt/β-catenin (cWnt) signaling is a crucial regulator of development and Dishevelled (Dsh/Dvl) functions as an integral part of this pathway by linking Wnt binding to the Frizzled:LRP5/6 receptor complex with β-catenin-stimulated gene expression. In many cell types Dsh has been localized to ill-defined cytoplasmic puncta, however in sea urchin eggs and embryos confocal fluorescence microscopy has shown that Dsh is localized to puncta present in a novel and development-essential vegetal cortex domain (VCD). In the present study, we used super-resolution light microscopy and platinum replica transmission electron microscopy (TEM) to provide the first views of the ultrastructural organization of Dsh within the sea urchin VCD. 3D structured illumination microscopy (SIM) imaging of isolated egg cortices demonstrated the graded distribution of Dsh in the VCD, whereas higher resolution stimulated emission depletion (STED) imaging revealed that some individual Dsh puncta consisted of more than one fluorescent source. Platinum replica immuno-TEM localization showed that Dsh puncta on the cytoplasmic face of the plasma membrane consisted of aggregates of pedestal-like structures each individually labeled with the C-terminus specific Dsh antibody. These aggregates were resistant to detergent extraction and treatment with drugs that disrupt actin filaments or inhibit myosin II contraction, and coexisted with the first cleavage actomyosin contractile ring. These results confirm and extend previous studies and reveal, for the first time in any cell type, the nanoscale organization of plasma membrane tethered Dsh. Our current working hypothesis is that these Dsh pedestals represent a prepositioned scaffold organization that is important for the localized activation of the cWnt pathway at the sea urchin vegetal pole. These observations in sea urchins may also be relevant to the submembranous Dsh puncta present in other eggs and embryos.

## Introduction

Wnt signaling controls a broad range of fundamental processes in cell and developmental biology including embryonic axis specification, and cell, tissue and organ morphogenesis and

**Funding:** Funding from collaborative National Science Foundation (www.nsf.gov) grants to J.H.H. (MCB-1917976) and C.B.S. (MCB-1917983). The funders had no role in study design, data collection and analysis, decision to publish, or preparation of the manuscript.

**Competing interests:** The authors have declared that no competing interests exist.

homeostasis [1]. Dishevelled (Dsh/Dvl) is a central integrator of the three main Wnt signaling pathways—the canonical Wnt/β-catenin pathway which drives cell fate specification, and the non-canonical Wnt/planar cell polarity and Wnt/Ca$^{2+}$ pathways controlling cellular morphogenesis [2–4]. Despite decades of research and clear evidence of the essential nature of Dsh in Wnt signal transduction in a number of species, fundamental questions remain about how Dsh is activated, regulated, and localized in cells [2, 5].

The sea urchin embryo has proved to be an exceptional experimental model for studying gene regulatory networks in general and how the canonical Wnt/β-catenin (cWnt) pathway regulates animal-vegetal axis determination in particular [6–8]. In the cWnt branch of this pathway Wnt ligand binding to the LRP5/6 and Frizzled (Fz) receptor complex activates Dsh which then inhibits the destruction complex comprised of Axin/APC/GSK-3β/CK1 α that targets β-catenin for proteasome-mediated degradation. The stabilization of β-catenin allows for it to accumulate first in the cytoplasm and then translocate into the nucleus where it acts as a transcription coactivator for a number of developmentally significant genes. In the sea urchin, localized activation of cWnt signaling with its associated β-catenin nuclearization in the vegetal blastomeres is a crucial determinant of endomesodermal specification and the patterning of the animal-vegetal axis [9–13]. The critical role for cWnt signaling in the early specification of the vegetal pole suggests that this region of the early sea urchin embryo may be enriched in Wnt ligands or receptors [14]. However, mRNA localization indicates that none of the maternally expressed Wnts or Fz receptors are preferentially concentrated in the vegetal pole region of the embryo [12, 13, 15, 16]. In addition, it has been argued that β-catenin nuclearization in the vegetal blastomeres is a cell autonomous process not under the influence of extracellular Wnt ligands given that nuclear localization of β-catenin occurs in putative vegetal blastomeres of dissociated embryos [10].

Interestingly, Dsh in sea urchin embryos has been shown to not only play it's expected role in the cWnt pathway regulation of the β-catenin-dependent gene expression of vegetal cells [12, 17], but also to localize in discrete puncta in a novel vegetal cortical domain (VCD) that arises in oocytes and persists in eggs and embryos [12, 14, 17, 18]. The VCD was initially recognized in early embryos overexpressing Dsh:GFP [17, 18] and subsequently shown to exist in oocytes, eggs and embryos up to the 60-cell stage using immunofluorescent localization [14]. Even though Dsh is maternally expressed uniformly in the egg and early embryo [16, 17], current evidence indicates that the VCD region is critical for its activation and the triggering of the cWnt pathway in the vegetal cells of the embryo. For example, overexpression of Dsh in zygotes had no impact on embryonic development whereas physical dissection of the VCD prior to fertilization resulted in abnormal, animalized/anteriorized embryos [12], an outcome also seen in embryos overexpressing dominant negative Dsh [17]. In addition, transplantation of the VCD to the animal pole induced the generation of ectopic endoderm [12]. All these results suggest that Dsh alone is not sufficient to direct proper cWnt-based development, but that instead it needs to be associated with the VCD which may act as a scaffold for localized Dsh and subsequent cWnt pathway activation [14].

Despite the developmental importance of the VCD, relatively little is known about the precise structural organization of the Dsh localized there. Past studies have shown that Dsh is arranged in cortex associated puncta and mutational analysis suggests that Dsh VCD binding is based on a N-terminal lipid-binding motif, the DIX domain, and a 21 amino acid motif between the Dsh PDZ and DEP domains [17, 18]. Previous work also suggests that while Dsh puncta are not sensitive to disruption of actin filaments or microtubules in the short term, longer term cytochalasin-based actin disruption led to the unexpected degradation of Dsh pools in the egg [14]. Poorly-defined punctate staining patterns, indications of membrane vesicle binding, and/or associations with the cytoskeleton have been reported for Dsh in cells from a

broad range of different species [2]. However, Dsh localization in cells remains controversial given that the punctate staining patterns have been considered non-physiological due to potential artifacts associated with biomolecular condensates formed due to Dsh over-expression [19], and other studies have suggested that Dsh can activate the cWnt pathway in the absence of cytoplasmic puncta [20].

In the present study we have localized sea urchin Dsh in the VCD at the ultrastructural level using super-resolution immunofluorescence microscopy and immuno-gold TEM in order to reveal the nanoscale architecture of plasma membrane tethered Dsh and investigate its potential relationship with cortical membrane structures and the actomyosin cytoskeleton. Our results indicate that Dsh in the sea urchin egg VCD is organized into well-defined puncta that consist of aggregates of multiple Dsh proteins that appear as groupings of pedestal-like structures interspersed between the microvillar cores of actin filaments in the cortex. These Dsh patches appear associated with the membrane but are not sensitive to detergent extraction suggesting an association with membrane proteins. We do not see evidence of a direct interaction between Dsh and cortical actin filaments given an analysis of our images and the persistence of Dsh puncta in eggs in which actin filaments have been disrupted with latrunculin treatment. These results confirm and extend previous studies and reveal, for the first time in any cell type, the ultrastructural organization of plasma membrane-tethered Dsh.

## Materials and methods

### Animals, antibodies, and reagents

*Lytechinus pictus* sea urchins were purchased from Marinus Scientific (Lakewood, CA) and *Strongylocentrotus purpuratus* sea urchins were collected from the waters surrounding Port Townsend, WA, and maintained at the Friday Harbor Laboratories (Friday Harbor, WA). All animals were kept in either running natural sea water or closed artificial sea water systems at 10–15˚C.

Primary antibodies used included anti-SUDsh-C, an affinity-purified rabbit polyclonal antibody raised against a synthetic peptide (NH$_2$-CMVPMMPRQLGSVPEDLSGS-COOH) based on a phylogenetically conserved sequence from the sea urchin Dsh protein C terminus, a mouse monoclonal antibody against a highly conserved epitope of chicken gizzard actin (clone C4) from EMD Millipore (Burlington, MA), and a mouse monoclonal antibody against the Ser19 phosphorylated form of the myosin II regulatory light chain (P-MyoRLC) from Cell Signaling Technology (Danvers, MA). Appropriate secondary antibodies conjugated to Alexa Fluor 488, 568, or Oregon Green as well as Alexa Fluor 633 conjugated phalloidin were obtained from Molecular Probes (Eugene, OR). Secondary antibody conjugated to 18 nm colloidal gold was obtained from Jackson ImmunoResearch Laboratories (West Grove, PA). The actin filament disruptor Latrunculin A (100 μg/ml stock in ethanol) and the myosin II light chain kinase (MLCK) inhibitor ML-7 (100 mM stock in DMSO) were obtained from Cayman Chemical (Ann Arbor, MI), whereas the fixable fluorescent membrane dye FM1-43FX (2 mM stock in methanol) was from Molecular Probes. Unless otherwise indicated, the majority of other reagents were purchased from either Sigma-Aldrich (St. Louis, MO) or Fisher Scientific (Pittsburgh, PA).

### Gamete collection, fertilization, cortex isolation, and inhibitor treatments

Sea urchin gametes were collected via intracoelomic injection with 0.5 M KCl, with sperm collected dry and eggs spawned in either natural sea water or MBL artificial sea water (ASW: 423 mM NaCl, 9 mM KCl, 9.27 mM CaCl$_2$, 22.94 mM MgCl$_2$, 25.5 mM MgSO$_4$, 2.14 mM NaHCO$_3$, pH 8.0) and subsequently dejellied by multiple washes with ASW. Eggs were

fertilized by addition of dilute sperm, the fertilization envelopes removed using 1 M urea (pH 8.0), and then washed into and reared in MBL calcium free sea water (CFSW: MBL ASW minus $CaCl_2$ and plus 1 mM EGTA) at 10–15˚C.

Cortices of unfertilized eggs and first division cycle embryos were generated as described in [21]. In brief, eggs/embryos were allowed to quickly settle onto poly-L-lysine (2 mg/ml) coated coverslips and then exposed to fluid shear force from a pipette containing an isotonic cortex isolation buffer (CIB: 0.8 M mannitol, 5 mM $MgCl_2$, 10 mM EGTA, 100 mM HEPES, pH 6.8 for unfertilized eggs and pH 7.4 for embryos). Isolated cortices were rinsed twice in CIB prior to further processing for light microscopic fluorescence localization.

In order to test the impact of detergent extraction on Dsh localization, isolated cortices were treated with 1% Triton X-100 in CIB for one minute following isolation and immediately prior to fixation. For disruption of actin filaments, eggs in ASW were treated for 20 min with 1 μM Latrunculin A. The effect of the inhibition of myosin II light chain kinase (MLCK) was tested by treating eggs in ASW for 20 min with 50 μM of the MLCK inhibitor ML-7.

## Fixation, fluorescent staining and light microscopic imaging and analysis

Isolated cortices plus and minus Triton extraction were fluorescently stained for membranes using 1–2 μM FM1-43 [22] for two minutes prior to fixation. Cortices were fixed in 2–4% form-aldehyde in CIB for 15 min followed by blocking in 2% goat serum and 1% BSA in PBS for 30 minutes. Immunostaining was performed with appropriate primary and secondary antibodies diluted in the range of 1:200 to 1:300 in blocking buffer and staining took place for 30–60 min for each stage. Fluorescent phalloidin was added to the secondary antibody staining step. Cortex samples for conventional and 3D-SIM microscopy were mounted in nonhardening Vectashield antifade mounting media (Vector Laboratories, Burlingame, CA), whereas STED imaging samples were mounted in Prolong Diamond mounting media (Molecular Probes).

Widefield epifluorescence microscopy of samples was performed on a Nikon (Tokyo, Japan) 80i microscope using either a 40X/0.75 NA Plan Fluor (phase contrast or DIC) or 60X/ 1.4 NA Plan Apo phase contrast objective lens with digital images captured using a Photo-metrics (Tuscon, AZ) CoolSnap Cf cooled CCD camera. Super-resolution microscopy was performed using two different methods. For 3D structured illumination microscopy (3D-SIM) [23], we utilized a DeltaVision OMX 3D-SIM Imaging System (GE Healthcare Bio-Sciences, Pittsburgh, PA) with an Olympus 60X/1.42 NA objective lens. Captured images were decon-volved and reconstructed using SoftWoRx software. Stimulated emission depletion (STED) super-resolution microscopy [24] was performed on a Leica (Wetzlar, Germany) Sp8 STED confocal using a 100X/1.4 NA objective lens.

All types of microscopic images were processed and analyzed using Fiji/ImageJ (Bethesda, MD). Graphs were prepared and statistical analysis carried out using GraphPad Prism 8 (San Diego, CA) with box and whisker plots having the following features: the box extends from the 25th to 75th percentiles; the whiskers extend to the minimum and maximum values; individual data points are plotted as red circles; the line in the middle of the box is the median. Final fig-ures were generated using Adobe Photoshop (San Jose, CA).

## Immuno-EM localization and platinum replica TEM

Immuno-EM localization of Dsh followed the methods of Svitkina [25, 26]. Isolated egg corti-ces were fixed with 0.25% glutaraldehyde in CIB for 5 min, rinsed with PBS and then quenched for 10 min in 2 mg/ml $NaBH_4$ in PBS. Following blocking in 2% goat serum, 1% BSA, and 1 mg/ml glycine in PBS for 30 min, cortices were incubated in primary anti-SUDsh-C antibody for 60 min followed by overnight incubation in colloidal gold-conjugated secondary antibody

in immuno-gold buffer (0.5 M NaCl, 20 mM Tris-HCl, pH 8.0, 0.05% Tween 20, 0.1% BSA). Following rinses in immuno-gold buffer, cortices were post fixed with 2% EM-grade glutaraldehyde in 0.1 M sodium cacodylate, pH 7.3 for 30 min.

The generation of critical point-dried and rotary-shadowed platinum replicas of immunolabled isolated cortices followed previously described methods [25–27]. Briefly, post fixation cortices were treated with aqueous 0.1% tannic acid followed by aqueous 0.2% uranyl acetate. Then the samples were dehydrated in a graded ethanol series, critical point dried, and rotary shadowed with platinum and carbon. The platinum replicas of cortices were separated from the glass coverslips using hydrofluoric acid, mounted on Formvar-coated grids, and observed on a JEM 1011 TEM (JEOL, Peabody, MA) operated at 100 kV and digital images captured with an ORIUS 832.10W CCD camera (Gatan Inc, Warrendale, PA) and presented in inverted contrast. All EM reagents and materials were obtained from Electron Microscopy Sciences (Hatfield, PA).

## Results and discussion

### The Dsh array in the VCD as visualized with super-resolution microscopy of isolated egg cortices

Super-resolution imaging using 3D-SIM showed that the Dsh localization pattern in the VCD of isolated egg cortices appeared similar to a dot diagram of a concentration gradient (Fig 1A–1H), as was previously documented using confocal imaging [14]. In the center of the VCD the density of Dsh puncta was high and then diminished when moving toward the edge of the distribution (Fig 1I). Measurements of the Dsh density per square micron (n = 15 cortices over 3 separate experiments) varied from an average of 3.6 in the center, to 1.8 midway in the distribution, to 0.7 in the sparse region of the edge (Fig 1I and 1J), with the differences in density between these three regions being statistically significant (p<0.001 based on a one-way ANOVA). 3D-SIM imaging (Fig 1A–1H) suggested that the Dsh puncta were discrete structures. However, imaging with higher resolution STED microscopy revealed that some of the Dsh puncta seen as individual fluorescent dots in confocal imaging (Fig 1M, arrows) were resolved as being composed of multiple spots in the STED images (Fig 1L, arrows). This indicated that the Dsh puncta consisted of aggregates of multiple Dsh proteins. The resolving power of STED is generally considered to be two-fold higher than SIM which is itself two-fold higher than conventional confocal imaging [28].

Within the VCD the Dsh puncta were interspersed with actin filaments that were present in two basic organizations: bright foci representing filament aggregations within the cores of the short microvillar found on the surface of unfertilized eggs [29, 30], and, particularly in *L. pictus* cortices, faint long actin filaments running parallel to the plane of the plasma membrane (Fig 1D arrow). The more elongate microvillar actin staining pattern seen in the *L. pictus* egg cortices (Fig 1A–1D) relative to the more circular microvilli in *S. purpuratus* cortices (Fig 1E–1H) are indicative of species-specific differences in microvillar morphology [30]. Our 3D-SIM images did not suggest a direct interaction between microvillar actin arrays and Dsh puncta, although in other cells and tissues it has been suggested that Dsh interacts with actin filaments, particularly within stress fibers and focal adhesions [31, 32].

### Platinum replica TEM of the egg VCD indicates that Dsh puncta consist of aggregates of pedestal-like structures

Immunogold labeling combined with platinum replica TEM of isolated egg cortices has allowed us to investigate the ultrastructure of the Dsh puncta visualized at the light

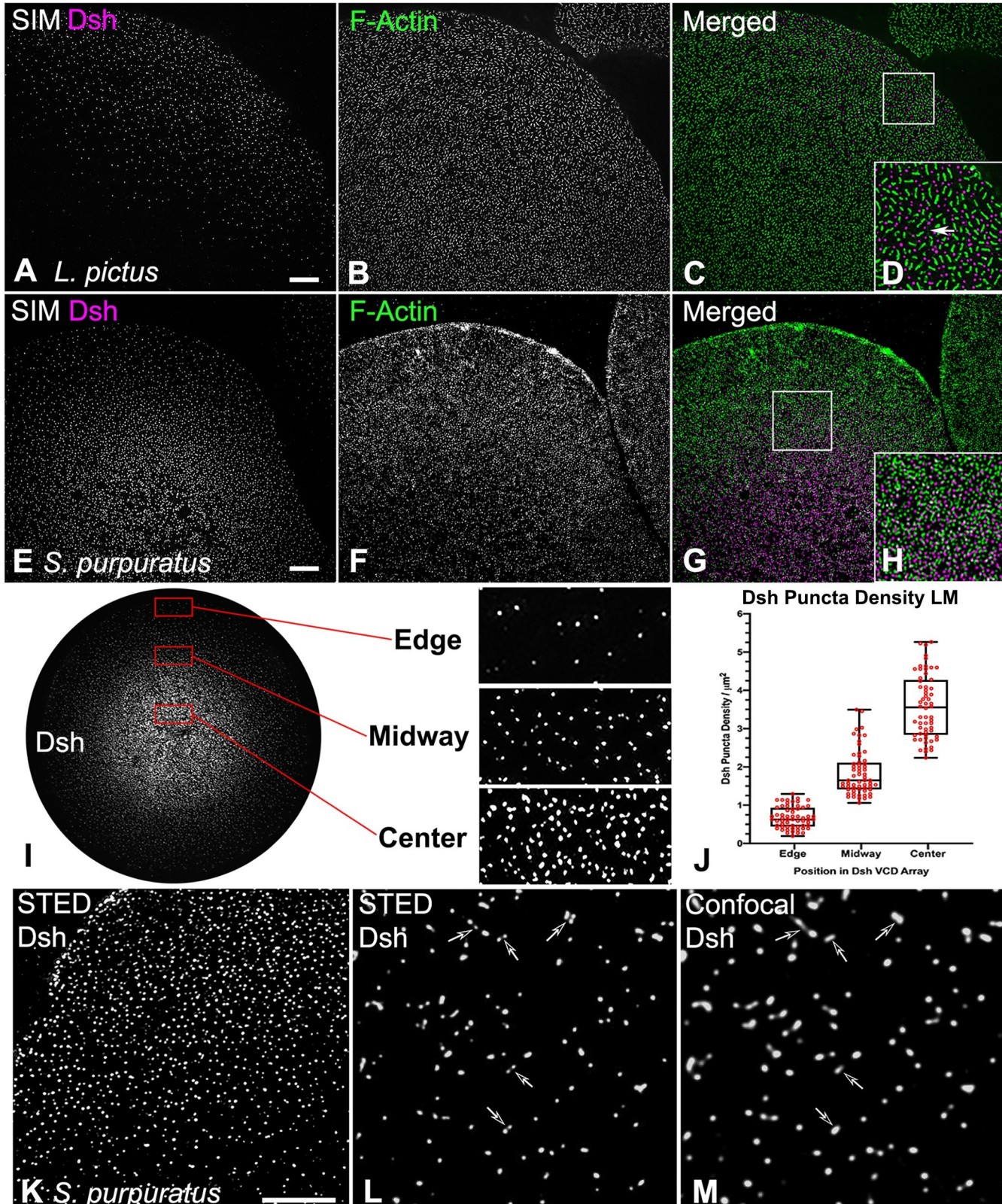

**Fig 1. Super-resolution imaging of Dsh and actin staining in the VCD in cortices isolated from unfertilized eggs.** (A-H) 3D-SIM imaging of Dsh (magenta) and F-actin (green) staining of cortices isolated from *L. pictus* (A-D) and *S. purpuratus* (E-H) eggs showing the distribution of Dsh puncta relative

to microvillar core actin foci. D and H are higher magnification versions of the 10x10 μm white boxes in C and G, and elongate submembranous actin filaments are visible in D (arrow). (I-M) The distribution of Dsh in the VCD appears similar to a diagrammatic representation of a concentration gradient. As shown with the enlarged versions of the 10x5 μm red boxes in I, the lowest density of Dsh puncta are at the VCD edge, with increased density in the midway point, and the maximum density in the center. (J) Quantification of Dsh densities in the three regions of the VCD. The Dsh VCDs of 5 cortices each from three separate experiments were analyzed and the densities in these 3 regions are all statistically significantly different. (K-M) STED imaging of Dsh staining in the *S. purpuratus* egg VCD reveals that single punctum in confocal images (L, arrows) often appear as multiple spots in the higher resolution STED images (M, arrows). Scale bars = 5 μm.

microscopic level. Low magnification TEM images of cortices (Fig 2A–2D) labeled for Dsh with 18 nm colloidal gold (colored gold in Figs 2B–2D and 3A–3F, 3I, 3J) demonstrated a sparse array of gold decorated aggregates (arrows in Fig 2A) amongst a distribution of micro-villar core (MV in Fig 2B) and submembranous actin filaments, as well as the remnants of membranous cortical granules (CG in Fig 2A), vesicles and endoplasmic reticulum. Higher magnification imaging (Fig 2E–2G) allowed for the identification of actin filaments (colored green in Fig 2E) due to their characteristic platinum replica TEM appearance [25–27, 33] and revealed that Dsh labeled aggregates appeared in well-defined patches on the cytoplasmic face of the plasma membrane (colored magenta in Fig 2E). A gallery of Dsh-labeled aggregates (Fig 3A–3F) showed that single colloidal gold particles were located on the top of pedestal-shaped structures that grouped together into patches that were distinct from the surrounding plasma membrane and underlying meshwork of the vitelline envelope. Within the aggregates the labeled structures often appeared close together and on occasion adopted linear or ring-shaped arrangements (Fig 3E and 3F). The Dsh aggregates did not colocalize with the tangles of short actin filaments present in microvillar cores (Figs 2 and 3), however on occasion they appeared adjacent to the submembranous elongate actin filaments running parallel with the plane of the membrane (Figs 2E, 2F and 3A, 3C, 3F, 3I). In terms of quantitative analysis of TEM images, the density of Dsh aggregates in the TEM images fell within the range of those seen with light microscopy (Fig 3G), the overall average area of single Dsh labeled aggregates was 16,600 nm$^2$ (Fig 3H; n = 6 cortices over 2 separate experiments), and there was no significant difference between the Dsh aggregate areas measured in the two species–*S. purpuratus* and *L. pictus* (Fig 3H). Comparison of regions of the same cortex that contained Dsh-labeled patches (Fig 3I) with areas that did not (Fig 3J) revealed that these unlabeled regions did not contain similar membrane-associated structures (Fig 3J), indicating that these structures were specific to the Dsh puncta in the VCD.

Weitzel et al. [17] and Leonard and Ettensohn [18] conducted mutational analysis of the association of sea urchin Dsh-GFP with the VCD. Their results indicated that this localization could be abolished by deletion of the entire N terminus, the DIX domain, a double mutation within DIX (K57A, E58A), or a 21 amino acid motif between the PDZ and DEP domains. These results combined with our TEM images suggest that Dsh is oriented with its N terminal domain closer to the membrane which may explain why our SUDshC-terminus specific anti-body appears to label the tops of the pedestal-like structures that are located farther away from the membrane surface. Dsh protein has also been shown to form oligomers via interactions between DIX domains in vitro and in vivo [34] and it is possible that the Dsh aggregates seen in our TEM images may represent some form of Dsh oligomerization in association with the membrane. A number of studies by Beinz and colleagues [34–37] have indicated that Dsh olig-omerization is critical for the formation of the cWnt signalosome and for increasing the affin-ity of binding to Axin that leads to the inactivation of the degradasomes that target β-catenin. In the sea urchin egg we have previously shown that the pool of Dsh in the VCD is differen-tially post-translationally modified compared to the Dsh not associated with the cortex [14]. Therefore, it is tempting to speculate that the Dsh aggregates that we have visualized corre-spond to a pre-positioned or "primed" cWnt signalosome precursor.

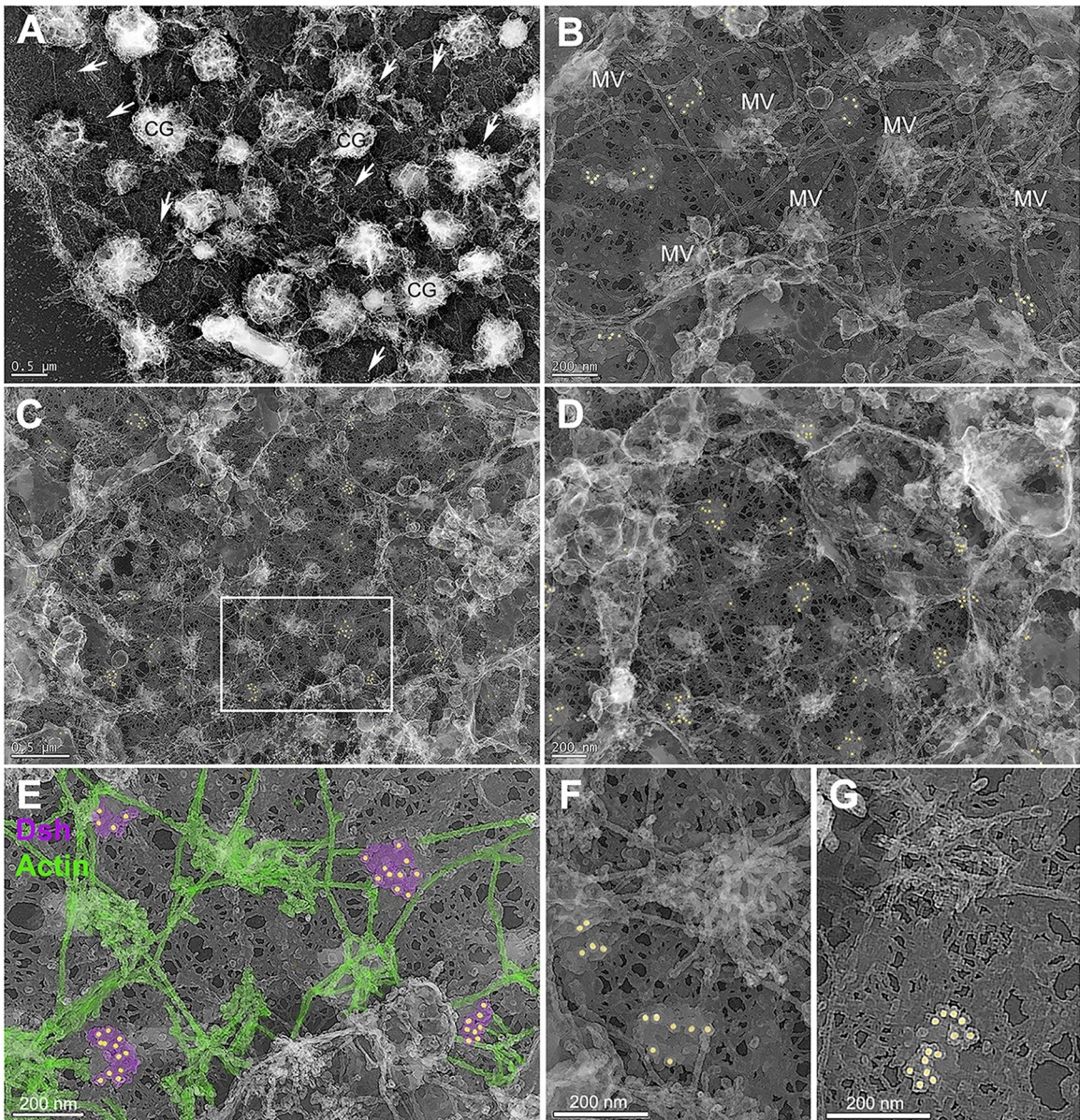

**Fig 2. Platinum replica TEM of Dsh and actin in the VCD in cortices isolated from unfertilized eggs.** Low (A,C) to medium (B,D) magnification images of cortices shows Dsh-specific colloidal gold (colored gold in B-G) staining of patches in the plane of the membrane (arrows in A). Tangled knots of short actin filaments appear in the cores of microvilli (MV in B) and numerous elongate actin filaments running parallel to the plane of the membrane are present. Cortical granules appear as shriveled structures (CG in A) and other membranous structures are also present. The meshwork that appears in the background of the images corresponds to the vitelline envelope. The white box in C appears at higher magnification in E in which Dsh aggregates are colored magenta and identifiable actin filaments in green. Dsh positive patches do not associate with MV core actin assemblages but do come in close proximity to submembranous actin filaments. (F,G) High magnification images indicate that Dsh patches consist of aggregates of pedestal-like structures—each labeled with a single colloidal gold particle—that can be grouped into one or more clusters. *L. pictus* cortices = A,B,F,G; *S. purpuratus* cortices = C,D,E. Bar length indicated in the images.

## The Dsh VCD array is resistant to Triton detergent extraction, disruption of actin filaments, and inhibition of myosin II contraction

The punctate Dsh staining present in sea urchin eggs and embryos and in many other cell types has been hypothesized to be caused by an association of Dsh with intracellular vesicles

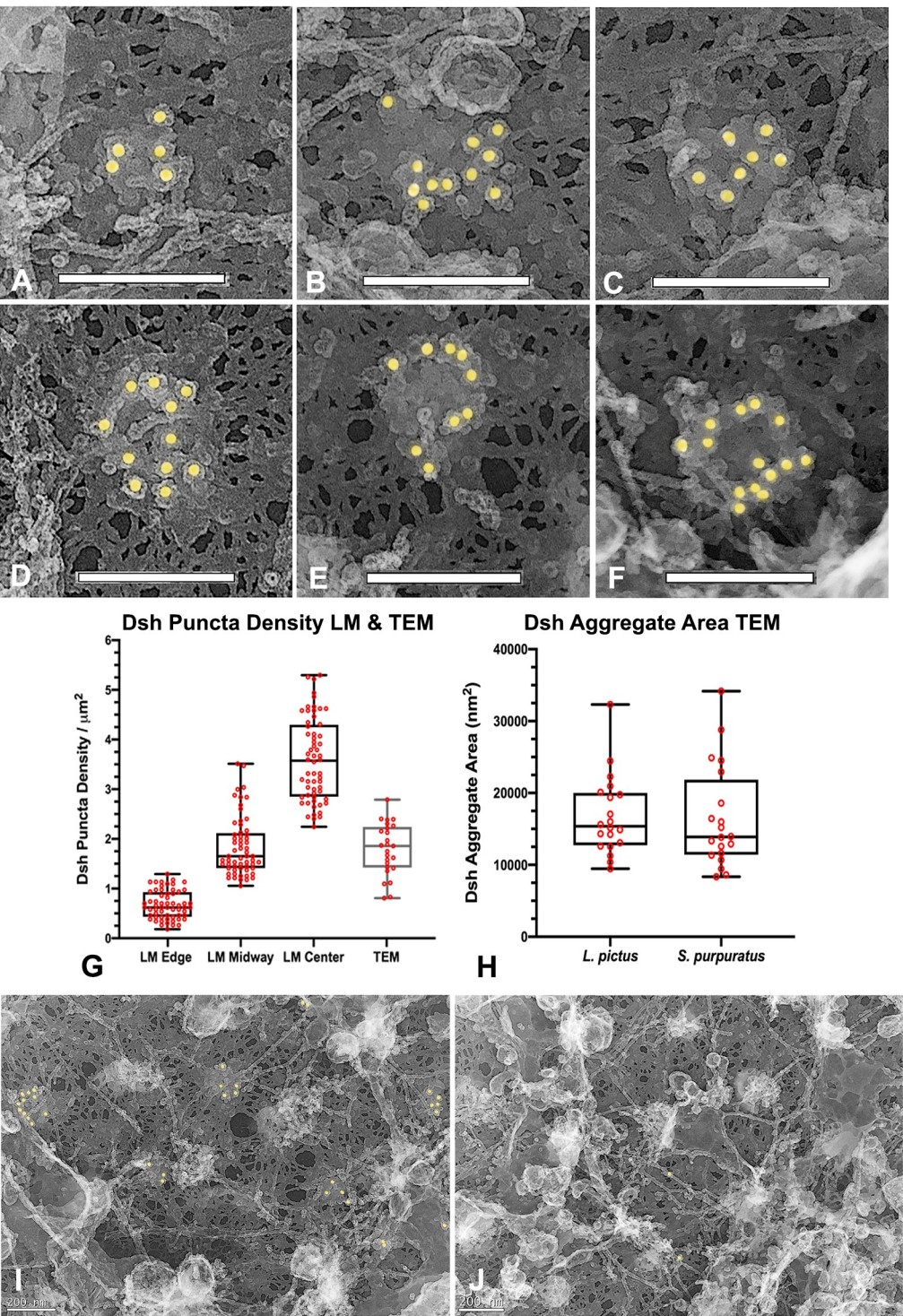

**Fig 3. Platinum replica TEM demonstrates that Dsh puncta correspond to patches of aggregates of pedestal-like structures.** (A-F) A gallery of high magnification images of Dsh labeled structures demonstrates that they consist of aggregates of pedestal-like structures in a variety of groupings. The top of each pedestal is labeled with a single colloidal gold particle suggesting that this area may correspond to the location of the C terminus of the Dsh protein. (G) The density of the Dsh puncta in TEM images from 3 separate cortices over three experiments (grey box) falls in the range of densities seen in the immunofluorescence images (black boxes reproduced from graph in Fig 1J). (H) The area of the Dsh aggregates from five cortices each from two separate experiments shows no statistically significant difference between the two species. (I,J) Comparison of regions of the same cortex in which Dsh labeling is present (I) and is not

present (J) shows that the patches associated with the Dsh labeling are not visible in the unlabeled regions, suggesting that these structures are specific to the Dsh puncta. *L. pictus* cortices = A,B,D,E; *S. purpuratus* cortices = C,F,I,J. Scale bars = 200 nm.

[2, 17, 18]. We tested the nature of the interaction between the Dsh puncta and membranes by performing a 1% Triton X-100 detergent extraction of egg cortices immediately after isolation and prior to fixation. Dsh immunofluorescent staining of the VCD persisted following detergent extraction (Fig 4A–4H) suggesting that the Dsh is in a detergent resistant structure associated with the plasma membrane. The disruption of membranes by the Triton-based extraction of cortices was confirmed by the absence of cortical granules in phase contrast images of extracted cortices (Fig 4D and 4H), as well as the loss of the membrane-dependent fluorescent staining with the lipophilic dye FM1-43 (Fig 4I–4N). These results suggest that Dsh interacts with either detergent resistant regions of the membrane–such as lipid rafts which have been previously identified in sea urchin eggs [38]–and/or it may be binding to a detergent resistant membrane protein assemblage. This lack of detergent sensitivity argues against the association of Dsh with the membranes of cortex-associated vesicles in the VCD, which is agreement with studies in other cells suggesting that Dsh puncta are not associated with cytoplasmic vesicles [39]. This includes work in *Xenopus* oocytes and embryos in which vegetal pole-associated Dsh puncta are argued to help direct cWnt-dependent axis specification following cortical rotation [40–42]. Tadjuidje et al. [42] have reported that *Xenopus* oocytes depleted of maternal Dsh (Dvl2/3) RNA still can activate cWnt signaling and that this derives from the persistence of Dsh puncta in the submembranous region of the vegetal cortex. These Dsh puncta did not codistribute with markers for endosomes, exocytotic vesicles or lysosomes and therefore do not appear associated with vesicles and may instead correspond to protein aggregates [42].

Our earlier work has indicated that the Dsh VCD localization in the sea urchin egg is sensitive to disruption of actin filaments using the drugs cytochalasin B and D and >2 hour incubation periods following a 20 min drug treatment [14]. In addition, studies in other cell types have argued for an interaction between Dsh and actin filaments [31, 32]. We tested the association between actin and Dsh in the short term using the drug Latrunculin A (LatA). Unlike the cytochalasins that tend to interfere with actin monomer addition to the plus end, LatA disrupts actin filaments primarily via monomer binding and subsequent sequestration. We treated embryos with 1 μM LatA for 20 min followed by isolation of cortices and anti-Dsh immunofluorescent staining. Dsh labeling of the VCD was maintained in LatA treated egg cortices even though the LatA led to a clear disruption of cortical actin filaments (Fig 5D–5F).

We also tested for the potential involvement of myosin II in the Dsh VCD localization given that the actomyosin cortex has been shown to influence the asymmetric distribution of the important polarity determining PAR protein complex in *C. elegans* [43, 44] and sea urchin [45] embryos. In sea urchins, the apical PAR localization was disrupted by treating embryos with the myosin II light chain kinase (MLCK) inhibitor ML-7 [45]. MLCK inhibition is known to block myosin II bipolar filament assembly and the enhancement of actin activated ATPase activity, and ML-7 has been used extensively as a myosin II inhibitor in previous studies on sea urchin embryos [45–48]. We treated eggs with 50 μM ML-7 for 20 min and then isolated the cortices and stained for Dsh. ML-7 treatment had no impact on Dsh VCD localization or actin distribution in the egg cortices (Fig 5G–5I).

## The Dsh VCD array is bisected by the cytokinetic contractile ring in isolated cortices of first cleavage embryos

The Dsh VCD array persists post-fertilization [14, 17, 18] despite the extensive remodeling of the plasma membrane caused by the mass exocytosis of cortical granules followed by the

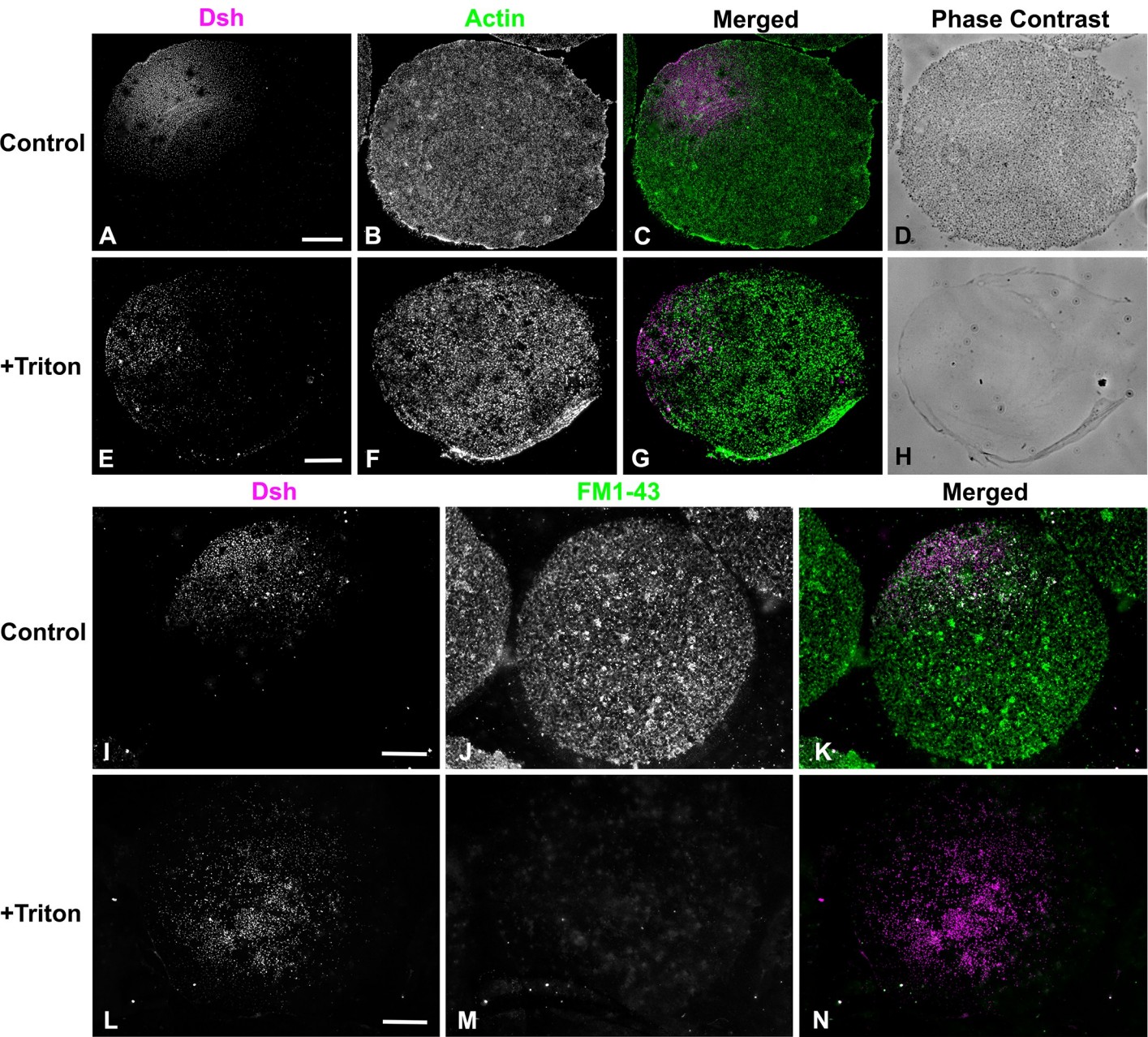

**Fig 4. The Dsh VCD array is resistant to Triton detergent extraction.** (A-H) Control unfertilized egg cortex from *L. pictus* (A-D) stained for Dsh (magenta) and actin (green) showing Dsh array and cortical granules in phase contrast. The Triton extracted cortex (E-H) demonstrates the persistence of the Dsh array following detergent extraction despite the loss of cortical granules seen in phase contrast. (I-N) Membrane staining with the fixable dye FM1-43 (green) shows that in control unfertilized egg cortices (I-K) that the Dsh (magenta) array is present along with a variety of membranous structures. In detergent extracted cortices (L-N) the Dsh array is still present even though specific membrane staining is lost. Scale bars = 10 μm.

endocytic uptake of membrane [49]. We investigated the behavior of the Dsh VCD array in early cleavage embryos by labeling cortices isolated from first division embryos with Dsh and probes specific for the cytokinetic contractile ring proteins myosin II and actin. We were interested in the relationship between the Dsh VCD array and the contractile ring given that our earlier work [27] has indicated that the contractile ring consists of a very dense assemblage of actin and myosin II filaments in close association with the plasma membrane. Previous work

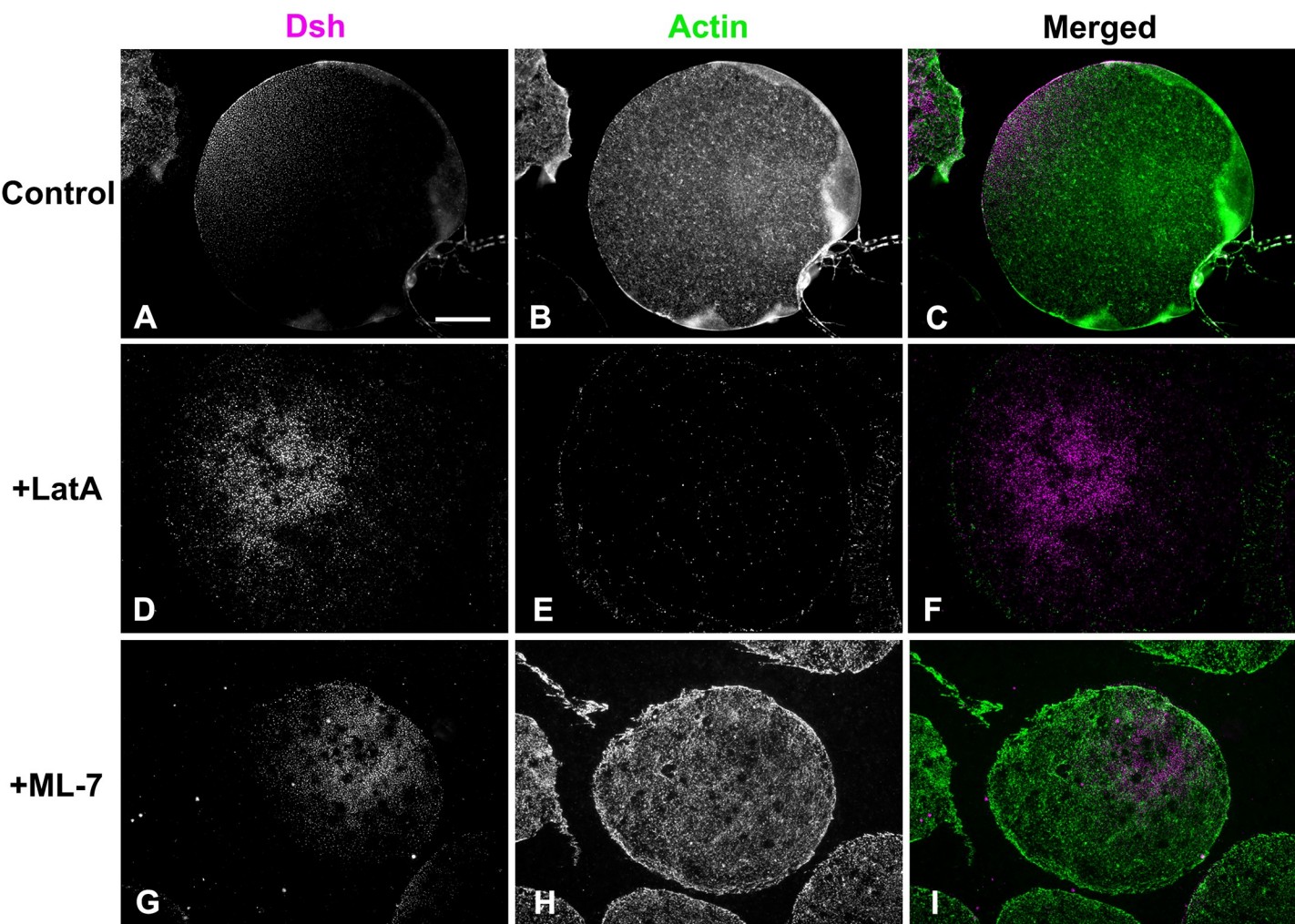

**Fig 5. The Dsh VCD persists following disruption of actin filaments or inhibition of myosin II contraction.** Control unfertilized egg cortex from *L. pictus* (A-C) contains the expected Dsh array (magenta) along with microvillar actin (green). In cortices from eggs treated with the actin filament disrupting drug LatA (D-F) the Dsh array persists (D) whereas the actin staining is greatly reduced (E). Both Dsh (G) and actin (H) staining appear unaffected in cortices isolated from unfertilized eggs treated with the MLCK inhibitor ML-7 (G-I) in order to inhibit myosin II contraction. Scale bar = 10 μm; magnifications of A-I are equivalent.

with either Dsh:GFP [17] or Dsh immunofluorescent labeling of whole embryos [14] has suggested that the first division cleavage furrow bisects the Dsh VCD. Cortices stained for Dsh and activated myosin II via staining with an antibody against the Ser19 phosphorylated version of the myosin II regulatory light chain (P-MyoRLC) revealed that the Dsh array coexisted with the contractile ring in regions in which they are both found (Figs 6 and 7). In the majority of isolated cleavage cortices, the Dsh distribution appeared to be bisected by the contractile ring structure (Figs 6 and 7), as would be expected for the two daughter cells to inherit roughly equivalent amounts of the VCD. In addition, the Dsh VCD array in first cleavage cortices differed from the VCD in unfertilized eggs in terms of the uniformity of the appearance and distribution of the Dsh puncta, as well as a significantly lower maximum density (unpaired t-test; $p < 0.0001$) in the central VCD (Fig 7; n = 6 UF and CL cortices each from 2 separate experiments). These changes in the Dsh puncta may be the result of the extensive restructuring of the cortex that takes place post fertilization and with the onset of first cleavage [49], and/or may be the result of Dsh puncta dynamics, to include potential coalescence.

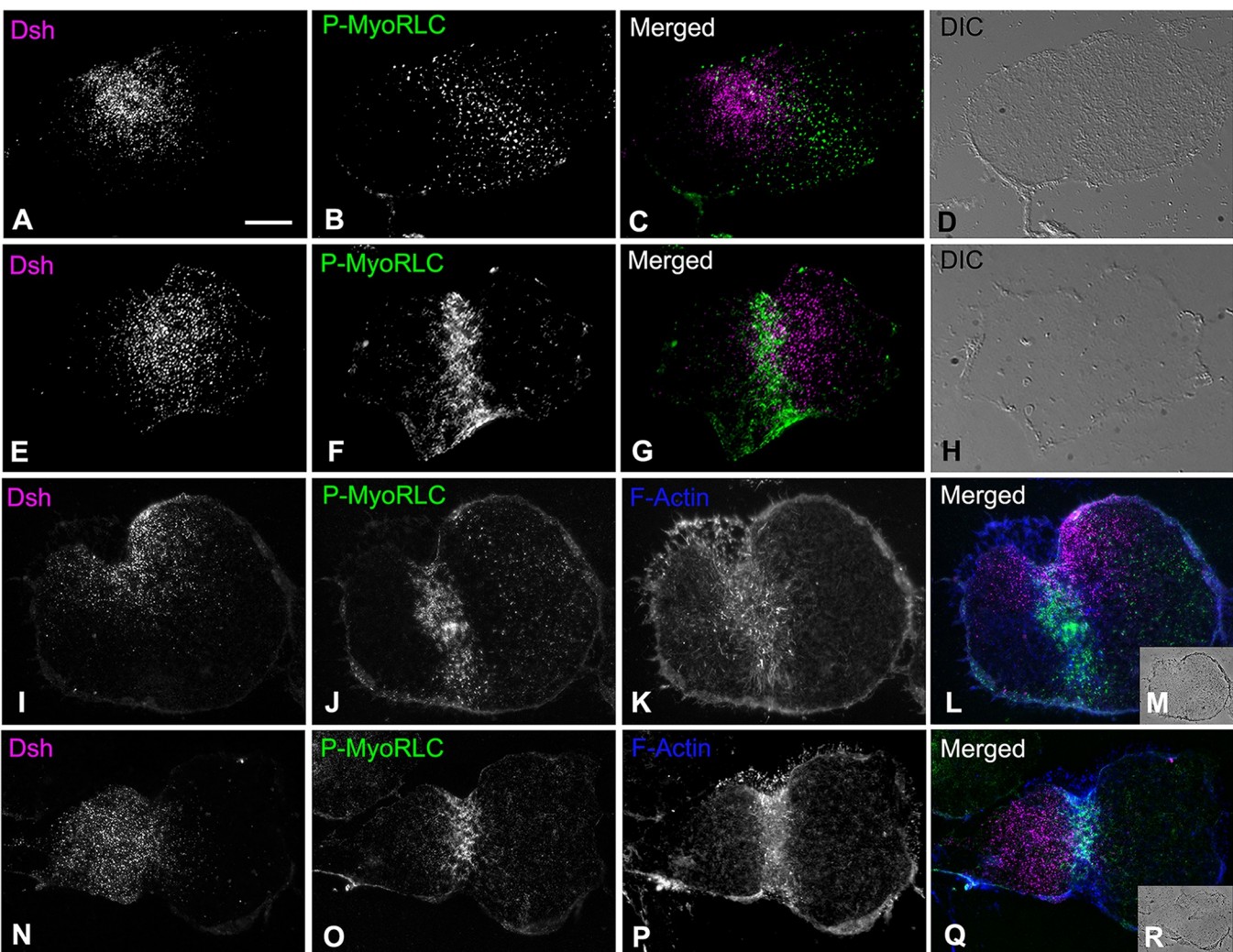

**Fig 6. The Dsh VCD in cortices isolated from first cleavage embryos is bisected by the cytokinetic contractile ring.** Widefield microscopy of cortices isolated from first division *S. purpuratus* embryos indicate that the Dsh punctate VCD array (A,E,I,N; magenta in C,G,L,Q) is often bisected by the cytokinetic contractile ring which labels for activated myosin II (P-MyoRLC; B,F,J,O; green in C,G,L,Q) and F-actin (K,P; blue in L,Q). Transmitted light DIC (D,H at equivalent magnification) and phase contrast (insets M,R at reduced magnification) images of individual cortices are provided for context. Scale bar = 10 μm, magnifications of A-L and N-Q are all equivalent.

## Conclusions

Despite some 25 years of research effort, aspects of the activation, regulation and localization of the central cWnt pathway protein Dsh remain enigmatic [2–4]. In the present study we use super-resolution and platinum replica transmission electron microscopy imaging of the sea urchin VCD to provide the first visualization of the nanoscale structural organization of membrane tethered Dsh. We show that Dsh is found associated with pedestal-like structures organized into aggregates located in the plane of the membrane, and that these clusters are resistant to detergent extraction and to treatment with inhibitors of actin polymerization and myosin II contraction. We also demonstrate that the Dsh VCD punctate arrays codistribute with the dense filamentous structure of the cytokinetic contractile ring in first division embryos. The general structural organization of membrane tethered Dsh that we have characterized in the sea urchin VCD may be applicable to the plasma membrane-associated Dsh

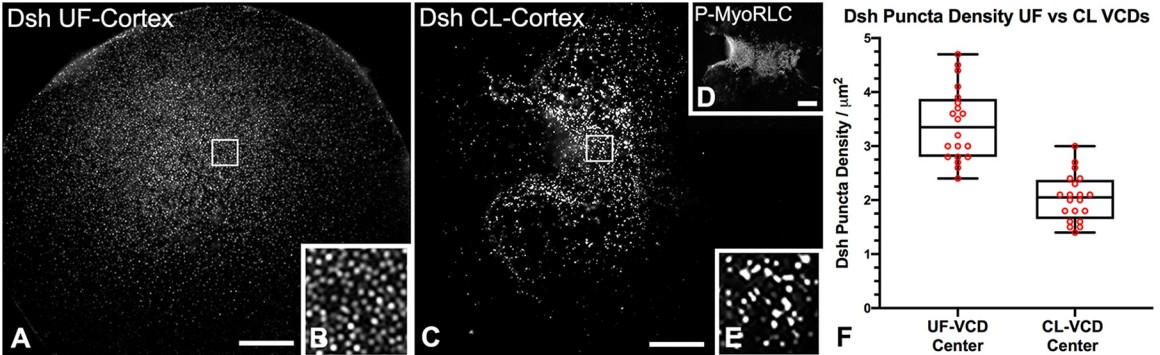

**Fig 7. Comparison of the Dsh VCD in cortices isolated from unfertilized eggs versus first cleavage embryos.** Widefield microscopy of the Dsh VCD in unfertilized (UF) egg (A,B) and first cleavage (CL) embryo (D,E) *S. purpuratus* cortices highlights the differences in the two arrays. Panels B and E correspond to higher magnification versions of the 5x5 µm white boxes in panels A and C. Panel D shows P-MyoRLC staining of the actomyosin contractile ring in the CL cortex seen at higher magnification in C. The maximum density of Dsh puncta in the central VCD in CL cortices is statistically less dense than in the central VCD of UF egg cortices (F), and the puncta appear not as uniform in structure or distribution (A,B,C,E). Scale bars = 10 µm.

puncta apparent in many cell types and particularly to those found in the vegetal cortex of *Xenopus* oocytes [40–42].

We speculate that the Dsh labeling in the sea urchin VCD could correspond to oligomers of Dsh present in the puncta and that our anti-SUDsh-C terminus antibody is labeling the end of the protein that is farthest from the surface of the membrane. Another intriguing possibility is that the Dsh aggregates might represent biomolecular condensates that have become associated with the membrane surface, as membranes have been argued to be one of the control centers of phase separation in a number of cell types [50] and condensates have been argued to play important roles in cWnt signaling [51]. The major Wnt pathway proteins Dsh, Axin, and APC all contain the condensate-requiring intrinsically disordered regions and have been shown to form dynamic, non-membrane enclosed puncta in cells, which suggests that the β-catenin destruction complex and the Dsh-dependent signalosome correspond to different forms of related biomolecular condensates [34, 51]. It is intriguing that in sea urchins [14] and in *Xenopus* [40] Dsh associated with vegetal puncta undergo postranslational modifications that correlate with cWnt activation. Our current working hypothesis is that the Dsh aggregates visualized in the present study correspond to a pre-positioned scaffold of a cWnt signalosome precursor that favors β-catenin signaling at the vegetal pole of the sea urchin embryo. Future studies will concentrate on establishing the localization and dynamics of other constituents of the cWnt signalosome, such as the sea urchin homologues of Fz, LRP5/6, and Axin.

## Supporting information

**S1 Table. Data used for graphs in Figs 1J, 3G, 3H and 7F plus statistical analysis.** (A) Shows density of Dsh puncta per square micron for LM regions of VCD (edge, midway and center) and TEM images used for graphs in Figs 1J and 3G. (B) One-way ANOVA analysis of Dsh puncta per square micron data for three LM regions of VCD shown in graph in Fig 1J. (C) Dsh aggregate area for TEM images of UF cortices from the two species which appears in graph in Fig 3H. (D) Results of t-test analysis of Dsh aggregate area data in panel C and Fig 3H. (E) Data for comparing Dsh puncta density in the central VCD in unfertilized eggs vs. first cleavage stage embryos shown in the graph in Fig 7F. (F) Results of t-test of analysis of Dsh punctate density data in panel E and Fig 7F.
(PDF)

**S1 Spreadsheet. Data used for graphs in Figs 1J, 3G, 3H and 7F.**
(XLSX)

## Acknowledgments

We gratefully acknowledge the expert assistance of Dr. Xufeng Wu (National Heart, Lung, and Blood Institute, National Institutes of Health) with 3D-SIM imaging, of Dr. Simon Watkins and Michael Calderon (Center for Biologic Imaging, University of Pittsburgh) with STED imaging, and Drs. Tanya Svitkina and Changsong Yang (University of Pennsylvania) with platinum replica TEM methods. Ethan Burg, Zoe Irons, Quenelle McKim and Hannah Herzon (Dickinson College) helped with experiments and/or image analysis. Thanks are also extended to Dr. Billie Swalla (University of Washington) for access to instrumentation and reagents while we were in summer residence at Friday Harbor Laboratories.

## Author Contributions

**Conceptualization:** John H. Henson, Charles B. Shuster, Athula H. Wikramanayake.

**Data curation:** John H. Henson, Bakary Samasa.

**Formal analysis:** John H. Henson, Bakary Samasa.

**Funding acquisition:** John H. Henson, Charles B. Shuster.

**Investigation:** John H. Henson, Bakary Samasa, Charles B. Shuster, Athula H. Wikramanayake.

**Methodology:** John H. Henson, Bakary Samasa, Charles B. Shuster.

**Project administration:** John H. Henson.

**Resources:** John H. Henson, Charles B. Shuster, Athula H. Wikramanayake.

**Supervision:** John H. Henson.

**Validation:** John H. Henson.

**Visualization:** John H. Henson.

**Writing – original draft:** John H. Henson, Charles B. Shuster, Athula H. Wikramanayake.

**Writing – review & editing:** John H. Henson, Charles B. Shuster, Athula H. Wikramanayake.

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
