## [Decision Letter · Decision Letter 0]

23 Mar 2021

PONE-D-21-05530

The nanoscale organization of the Wnt signaling integrator Dishevelled in the development-essential vegetal cortex domain of an egg and early embryo

PLOS ONE

Dear Dr. Henson,

Thank you for submitting your manuscript to PLOS ONE. After careful consideration, we feel that it has merit but does not fully meet PLOS ONE’s publication criteria as it currently stands. Therefore, we invite you to submit a revised version of the manuscript that addresses the points raised during the review process.

Please addressed all of the reviewers comments point by point.  I think  I agree with the second reviewer that title might be better - perhaps something like "**Nanoscale organization of the Wnt signaling integrator Dishevelled in the vegetal domain of the sea urchin egg and early embryo".  **But I leave that to you.

We look forward to receiving your revised manuscript.

Kind regards,

Michael Klymkowsky, Ph.D.

Academic Editor

PLOS ONE

Journal Requirements:

Reviewers' comments:

Reviewer's Responses to Questions

**Comments to the Author**

1. Is the manuscript technically sound, and do the data support the conclusions?

Reviewer #1: Yes

Reviewer #2: Yes

2. Has the statistical analysis been performed appropriately and rigorously? 

Reviewer #1: N/A

Reviewer #2: Yes

3. Have the authors made all data underlying the findings in their manuscript fully available?

Reviewer #1: Yes

Reviewer #2: Yes

4. Is the manuscript presented in an intelligible fashion and written in standard English?

Reviewer #1: Yes

Reviewer #2: Yes

5. Review Comments to the Author

Reviewer #1: The authors of "The nanoscale organization of the Wnt signaling integrator Dishevelled in the development-essential vegetal cortex domain of an egg and early embryo" offer a beautiful set of images showing ever more clearly the localisation of Dvl in the VCD. The transition form SIM to STED to TEM is particularly effective. There are a couple of issues that should be addressed.

1. Looking at the references cited, and the general discussion of Dvl puncta, the authors do not address the current view of signalling activation through a transition from degradation complexes to activation signalosomes as most recently detailed by work from the Bienz lab and others. The working hypothesis at the end of the Conclusion section is very vague.

2. From the images: Figure 2, the text states "colored gold in Fig 2A-D, but this color was difficult to see in the image.

3. The authors focus on canonical signaling, but the images in fig 2 show some Dsh puncta at the ends of actin filaments so could this not also suggest polarity signaling?

4. In figure 6, are the colors not reversed in panel L? Again here, the correlation is between polarity and not canonical signaling, or did I misunderstand the intent.

Overall, the focus on Dvl localization is informative, but Dvl is central to the canonical pathway and to the non-canonical pathways so unless downstream activity is monitored, it isn't obvious why the authors say this is a canonical pathway effect.

Reviewer #2: This is a very valuable study that provides the first high-resolution imaging of Dsh-associate structures in the sea urchin egg, with implications for the organization of Dsh in many other animal oocytes and early embryos. The images, even the low-resolution versions in the PDF provided for review, are striking. The study reveals that Dsh puncta (as defined by conventional epifluorescence imaging) are composed of several sub-puncta that sit on pedestal-like structures. It clarifies the relation between Dsh puncta and a) cortical/microvillar actin and b) detergent-soluble membranes.

I believe the paper could be strengthened in two ways, although I don’t consider these pre-requisites for publication:

1) Since the result with ML-7 is negative (i.e., it has no effect on Dsh localization), there need to be some positive controls to indicate that the drug is effective at inhibiting myosin II at the concentrations used and in this species.

2) The authors suggest that the Dsh VCD array in first division cortices is not as highly ordered as in egg cortices- they say (L406-408) that the VCD “often does not closely resemble a concentration gradient pattern and the individual puncta appear larger and less densely distributed (Fig 6).”. It would be valuable to support these anecdotal statements with real measurements. I’m also curious as to how the authors interpret such changes- are they suggesting that Dsh puncta disassemble after fertilization, perhaps in a non-random way? How could a graded distribution be converted into a non-graded distribution?

Minor suggestions:

1) The title is clumsy- I would remove “development essential”

2) In the abstract, define definitions (SIM, STED, TEM)

3) In the abstract- “the concentration gradient-like distribution” is clumsy- just say “graded distribution”

4) L91 “Even though Dsh is maternally expressed uniformly in the egg and early embryo [16,17]” Change “Dsh” to “Dsh mRNA”

5) Clarify stages shown in some of the figures. “Egg” can mean different things to different people so better to say “unfertilized egg” if that’s what is shown.

6) The authors refer to the K57A, E58A as affecting a phospholipid-binding region of DIX but this was originally based on the old Overduin 2002 study and I believe since then data from the Bienz lab have instead implicated these mutations in DIX oligomerization.

7) Reference list- the Tadjuidje reference is missing a few words

8) L444- what is meant by an “autonomous” scaffold?

6. PLOS authors have the option to publish the peer review history of their article (what does this mean?). If published, this will include your full peer review and any attached files.

Reviewer #1: No

Reviewer #2: No

---

## [Author Response · Author response to Decision Letter 0]

4 May 2021

Response to Reviewers

First, we would like to thank the reviewers for their constructive comments and emphasize that we have made every effort to address their concerns. Our point by point responses to their reviews appear below. Second, our revision includes some additional quantitative analysis of the images represented by the graphs in Figs 1J, 3G and 3H. Additional measurements of TEM images were performed for both density (Fig 3G) and aggregate area (Fig 3H) measurements, with the area measurements recategorized by individual species. This emphasized the lack of significant difference in the areas of the Dsh aggregates that appear in L. pictus and S. purpuratus egg cortices. In the interest of data transparency, we also decided to show all the data points (as red circles) in the graphs in Figs 1J, 3G, 3H and 7F, as well as submit supplementary information files (S1 Table and S2 Spreadsheet) showing all the data used to build the graphs and statistical analyses. 

Reviewer #1: The authors of "The nanoscale organization of the Wnt signaling integrator Dishevelled in the development-essential vegetal cortex domain of an egg and early embryo" offer a beautiful set of images showing ever more clearly the localisation of Dvl in the VCD. The transition from SIM to STED to TEM is particularly effective. There are a couple of issues that should be addressed.

1. Looking at the references cited, and the general discussion of Dvl puncta, the authors do not address the current view of signaling activation through a transition from degradation complexes to activation signalosomes as most recently detailed by work from the Bienz lab and others. The working hypothesis at the end of the Conclusion section is very vague.

We’ve addressed the first suggestion referring to the work by Bienz and colleagues by expanding a section of the Results and Discussion on page 10, lines 323-326 that originally referred to the possible role of Dsh oligomers reviewed in Gammons and Beinz (2018). This new section references 4 studies by Bienz and coworkers (2014; 2016; 2018; 2020) that highlight the critical role of Dsh oligomerization in the formation of the Wnt signalosome and how this structure recruits Axin in order to inhibit the -catenin destruction complexes. Below is this new section (italicized) found on page 10, lines 329-335 in the revised manuscript:

“Dsh protein has also been shown to form oligomers via interactions between DIX domains in vitro and in vivo [34] and it is possible that the Dsh aggregates seen in our TEM images may represent some form of Dsh oligomerization in association with the membrane. A number of studies by Beinz and colleagues [34-37] have indicated that Dsh oligomerization is critical for the formation of the cWnt signalosome and for increasing the affinity of binding to Axin that leads to the inactivation of the degradasomes that target -catenin. In the sea urchin egg we have previously shown that the pool of Dsh in the VCD is differentially post-translationally modified compared to the pool of Dsh not associated with the cortex [14]. Therefore, it is tempting to speculate that the Dsh aggregates that we have visualized correspond to a pre-positioned or “primed” cWnt signalosome precursor.”

We’ve addressed the second comment about the vague nature of the working hypothesis in the Conclusions section by rewriting this sentence (page 14, lines 467-471) as follows:

“Our current working hypothesis is that the Dsh aggregates visualized in the present study correspond to a pre-positioned scaffold of a cWnt signalosome precursor that favors -catenin signaling at the vegetal pole of the embryo. Future studies will concentrate on establishing the localization and dynamics of other constituents of the cWnt signalosome, such as the sea urchin homologues of Fz, LRP5/6, and Axin.”

2. From the images: Figure 2, the text states "colored gold in Fig 2A-D”, but this color was difficult to see in the image. 

The lower resolution images in the reviewer’s version of the figures may have made it difficult to discern the colored colloidal gold particles, especially in the lower magnification images present in panels A-D. In panel A the magnification is sufficiently low that the gold particles are too small to be colored so we now refer to gold colored particles only in panels B-G in the Fig 2 legend. Readers of the higher resolution published version of this paper will have the ability to zoom in on this figure in order to more easily visualize the colored particles. 

3. The authors focus on canonical signaling, but the images in Fig 2 show some Dsh puncta at the ends of actin filaments so could this not also suggest polarity signaling?

We can’t rule out an association between actin filaments and Dsh in the VCD, however our TEM images by and large don’t demonstrate a close association between the two. In addition, Fig 5D-F shows that Dsh staining remains in cortices in which F-actin staining has been lost due to treatment with Latrunculin A. Our previous published work (Peng and Wikramanayake, 2013) also demonstrated that Dsh puncta remained - at least for 2 hours - in whole eggs treated with the actin filament disrupting drugs cytochalasin B and D. 

4. In figure 6, are the colors not reversed in panel L? Again here, the correlation is between polarity and not canonical signaling, or did I misunderstand the intent.

The colors of Dsh and P-MyoRLC were indeed incorrectly assigned in Fig 6L and we have corrected them in the revised version of the figure. 

In Fig 6 we are highlighting the persistence of the Dsh VCD array even in the midst of the dense contractile ring structure that we have previously defined (Henson et al., 2017), and that the Dsh array is typically split between the two blastomeres during first division. Our previous work (Peng and Wikarmanayake, 2013) has indicated that as the embryo continues to divide that the Dsh VCD remains at the vegetal pole which by the 60-cell stage corresponds to the cells that show the cWnt pathway characteristic of nuclear localization of -catenin. 

Overall, the focus on Dvl localization is informative, but Dvl is central to the canonical pathway and to the non-canonical pathways so unless downstream activity is monitored, it isn't obvious why the authors say this is a canonical pathway effect.

As we have indicated in the Introduction section of the manuscript, the early sea urchin embryo is a model system for studying how the cWnt pathway regulates AV/AP axis determination via the activation of -catenin regulated gene expression. -catenin nuclearization in the vegetal blastomeres of the 60-cell late blastula/early gastrula embryo is a crucial determinant of endomesodermal specification and the maturation of the AV/AP axis. The Dsh VCD that we characterize in the present study marks the vegetal pole of the egg and early embryo and as such serves as a predeterminant for cWnt signaling-based -catenin stabilization. Localized determinants of -catenin stabilization such as the Dsh VCD in the sea urchin are characteristic of vegetal/posterior pole determination in all invertebrate deuterostome embryos that have been studied to date. Therefore, although we can’t absolutely negate the possibility that the sea urchin Dsh VCD plays some yet to be defined role in ncWnt signaling, we are much more confident of its role in cWnt signaling and have concentrated on that. 

Reviewer #2: This is a very valuable study that provides the first high-resolution imaging of Dsh-associate structures in the sea urchin egg, with implications for the organization of Dsh in many other animal oocytes and early embryos. The images, even the low-resolution versions in the PDF provided for review, are striking. The study reveals that Dsh puncta (as defined by conventional epifluorescence imaging) are composed of several sub-puncta that sit on pedestal-like structures. It clarifies the relation between Dsh puncta and a) cortical/microvillar actin and b) detergent-soluble membranes.

I believe the paper could be strengthened in two ways, although I don’t consider these pre-requisites for publication:

1) Since the result with ML-7 is negative (i.e., it has no effect on Dsh localization), there need to be some positive controls to indicate that the drug is effective at inhibiting myosin II at the concentrations used and in this species.

Our previous published work using 50 µM of ML-7 to treat L. pictus embryos (Lucero et al., 2006. Mol. Biol. Cell 17:4093-4103; reference #47 in the manuscript) has shown that this concentration of the drug completely inhibits the myosin II-based increase in cortical contractility that accompanies the first division metaphase-anaphase transition, as well as cytokinesis. 

2) The authors suggest that the Dsh VCD array in first division cortices is not as highly ordered as in egg cortices- they say (L406-408) that the VCD “often does not closely resemble a concentration gradient pattern and the individual puncta appear larger and less densely distributed (Fig 6).”. It would be valuable to support these anecdotal statements with real measurements. I’m also curious as to how the authors interpret such changes- are they suggesting that Dsh puncta disassemble after fertilization, perhaps in a non-random way? How could a graded distribution be converted into a non-graded distribution?

As a result of new experimentation, we are able to at least partially address the reviewer’s request that we provide additional information about the differences between the Dsh VCD in cortices isolated from unfertilized eggs versus first cleavage embryos. In a new Figure 7 (plus legend) we provide a direct comparison between a Dsh VCD in an egg (7A,B) and one in a first division embryo (7C,E) at equivalent low (7A,C) and high (7B,E) magnifications. In Figure 7F we show a graph of the density of Dsh puncta in the center of the VCD in egg and cleaving embryo cortices which demonstrates a statistically significantly higher density of Dsh puncta in the egg cortex. We also added a new section that speculates about how the Dsh VCD is changing which appears near the top of page 13 and is reproduced in italics below. However, we think that our current understanding of Dsh dynamics in the early embryo is too preliminary to address whether Dsh puncta disassemble or how the VCD distributions might be transformed. This must await the results of future experiments concentrating on the live cell imaging of Dsh during embryogenesis. 

“In addition, the Dsh VCD array in first cleavage cortices differed from the VCD in unfertilized eggs in terms of the uniformity of the appearance and distribution of the Dsh puncta, as well as a significantly lower maximum density (unpaired t-test; p<0.0001) in the central VCD (Fig 7; n = 8 UF and CL cortices each from 2 separate experiments). These changes in the Dsh puncta may be the result of the extensive restructuring of the cortex that takes place post fertilization and with the onset of first cleavage [49], and/or may be the result of Dsh puncta dynamics, to include potential coalescence.”

Minor suggestions:

1) The title is clumsy- I would remove “development essential”.

This has been removed. The new title is “Nanoscale organization of the Wnt signaling integrator Dishevelled in the vegetal cortex domain of an egg and early embryo”. We decided to leave sea urchin out of the title in the hope that this would stimulate more general interest in the paper, given that vegetal pole concentrations of Dsh are not exclusive to sea urchin eggs/embryos. 

2) In the abstract, define definitions (SIM, STED, TEM).

These abbreviations/acronyms have been defined in the abstract. 

3) In the abstract- “the concentration gradient-like distribution” is clumsy- just say “graded distribution”.

This has been changed. 

4) L91 “Even though Dsh is maternally expressed uniformly in the egg and early embryo [16,17]” Change “Dsh” to “Dsh mRNA”.

This has been changed. 

5) Clarify stages shown in some of the figures. “Egg” can mean different things to different people so better to say “unfertilized egg” if that’s what is shown.

“Unfertilized egg” has been added to the figure legends of Figs 1, 2, 4, and 5. 

6) The authors refer to the K57A, E58A as affecting a phospholipid-binding region of DIX but this was originally based on the old Overduin 2002 study and I believe since then data from the Bienz lab have instead implicated these mutations in DIX oligomerization.

On page 10 we deleted the phrase that identified the K57A, E58A double mutant as being a phospholipid binding region of DIX. The current sentence reads as follows:

“Their results indicated that this localization could be abolished by deletion of the entire N terminus, the DIX domain, a double mutation within DIX (K57A, E58A), or a 21 amino acid motif between the PDZ and DEP domains.”

7) Reference list- the Tadjuidje reference is missing a few words.

This reference title has been corrected to read: “The functions of maternal Dishevelled 2 and 3 in the early Xenopus embryo.”

8) L444- what is meant by an “autonomous” scaffold?

The word “autonomous” has been deleted in this sentence and the sentence has been rewritten in response to a suggestion from another reviewer. It now reads: “Our current working hypothesis is that the Dsh aggregates visualized in the present study correspond to a pre-positioned scaffold of cWnt signalosome precursor that favors -catenin signaling at the vegetal pole of the sea urchin embryo.”

---

## [Decision Letter · Decision Letter 1]

12 May 2021

The nanoscale organization of the Wnt signaling integrator Dishevelled in the vegetal cortex domain of an egg and early embryo

PONE-D-21-05530R1

Dear Dr. Henson,

We’re pleased to inform you that your manuscript has been judged scientifically suitable for publication and will be formally accepted for publication once it meets all outstanding technical requirements.

Kind regards,

Michael Klymkowsky, Ph.D.

Academic Editor

PLOS ONE

Additional Editor Comments (optional):

Reviewers' comments:

Reviewer's Responses to Questions

**Comments to the Author**

1. If the authors have adequately addressed your comments raised in a previous round of review and you feel that this manuscript is now acceptable for publication, you may indicate that here to bypass the “Comments to the Author” section, enter your conflict of interest statement in the “Confidential to Editor” section, and submit your "Accept" recommendation.

Reviewer #1: All comments have been addressed

Reviewer #2: All comments have been addressed

2. Is the manuscript technically sound, and do the data support the conclusions?

Reviewer #1: Yes

Reviewer #2: (No Response)

3. Has the statistical analysis been performed appropriately and rigorously? 

Reviewer #1: Yes

Reviewer #2: (No Response)

4. Have the authors made all data underlying the findings in their manuscript fully available?

Reviewer #1: Yes

Reviewer #2: (No Response)

5. Is the manuscript presented in an intelligible fashion and written in standard English?

Reviewer #1: Yes

Reviewer #2: (No Response)

6. Review Comments to the Author

Reviewer #1: The reviewers' comments have been addressed and the manuscript can now be published. The data is well presented and of interest to the field.

Reviewer #2: (No Response)

7. PLOS authors have the option to publish the peer review history of their article (what does this mean?). If published, this will include your full peer review and any attached files.

Reviewer #1: No

Reviewer #2: No

---

## [Editor Report · Acceptance letter]

17 May 2021

PONE-D-21-05530R1 

The nanoscale organization of the Wnt signaling integrator Dishevelled in the vegetal cortex domain of an egg and early embryo 

Dear Dr. Henson:

I'm pleased to inform you that your manuscript has been deemed suitable for publication in PLOS ONE. Congratulations! Your manuscript is now with our production department. 

Kind regards, 

on behalf of

Dr. Michael Klymkowsky 

Academic Editor

PLOS ONE